# Menstrual health intervention and school attendance in Uganda (MENISCUS-2): a pilot intervention study

Catherine Kansiime,[1] Laura Hytti,[2] Ruth Nalugya,[1] Kevin Nakuya,[1] Prossy Namirembe,[1] Shamirah Nakalema,[2] Stella Neema,[3] Clare Tanton,[4] Connie Alezuyo,[5] Saidat Namuli Musoke,[6] Belen Torondel,[7] Suzanna C Francis,[8] David A Ross,[9] Christopher Bonell [iD],[10] Janet Seeley,[1,10] Helen Anne Weiss [iD] [8]

For numbered affiliations see end of article.

**Correspondence to**
Dr Helen Anne Weiss;
helen.weiss@lshtm.ac.uk

## ABSTRACT

**Objectives** Achieving good menstrual health and hygiene (MHH) is a public health challenge and there is little evidence to inform interventions. The aim of this study was to pilot test an intervention to improve MHH and school attendance in Uganda, in preparation for a future cluster-randomised trial.

**Design** Longitudinal study with pre–post evaluation of a pilot intervention.

**Setting** Two secondary schools in Entebbe, Uganda.

**Participants** Of the 473 eligible students in secondary 2 (S2) at baseline, 450 (95.1%; 232 girls and 218 boys) consented/assented. 369 students (188 girls; 81.0%; and 181 boys; 83.0%) participated in the endline survey.

**Intervention** The intervention comprised training teachers to improve delivery of government guidelines for puberty education, training in use of a menstrual kit and pain management, a drama skit, provision of analgesics and improvements to school water and sanitation hygiene facilities.

**Primary and secondary outcome measures** Feasibility and acceptability of delivering the intervention. Baseline and endline quantitative surveys were conducted, with qualitative interviews conducted at endline. School attendance was assessed using self-completed daily diaries among a nested cohort of 100 female students.

**Results** There were high levels of uptake of the individual and behavioural intervention components (puberty education, drama skit, menstrual hygiene management (MHM) kit and pain management). The proportion of girls reporting anxiety about next period decreased from 58.6% to 34.4%, and reported use of effective pain management increased from 76.4% to 91.4%. Most girls (81.4%) reported improved school toilet facilities, which improved their comfort managing menstruation. The diary data and qualitative data indicated a potential intervention impact on improving menstrual-related school absenteeism.

**Conclusions** The pilot study showed that the multicomponent MHM intervention was acceptable and feasible to deliver, and potentially effective in improving menstruation knowledge and management. A cluster-randomised trial is needed to evaluate rigorously the intervention effects on MHM and school attendance.

### Strengths and limitations of this study

► The study evaluated a multicomponent menstrual health intervention that addresses individual, behavioural and environmental barriers to good menstrual health and school attendance.
► The study population are students in periurban secondary schools, which is important given the recognised importance of girls' secondary education to future development, and evidence that school absenteeism due to menstruation is problematic for secondary school girls.
► This is the first menstrual health intervention to address menstrual pain—a major contributor to school absence in girls.
► The intervention includes boys as well as girls, as sustainable changes in menstrual management depend on addressing stigma about menstruation.
► The conclusions are limited due to the lack of a control group, which means that the improvements seen in knowledge, school attendance and wellbeing may reflect the girls being older at endline or differences in attendance in different terms.

**Trial registration number** NCT04064736; Pre-results.

## INTRODUCTION

Many girls and women globally lack the knowledge, materials and facilities for safely managing menstruation without stigma.[1–4] Adequate menstrual hygiene management (MHM) is defined as (1) using clean menstrual management material to absorb or collect menstrual blood, (2) that can be changed in privacy as often as necessary for the duration of a menstrual period, (3) using soap and water for washing the body as required, (4) having access to safe and convenient facilities to dispose of used menstrual management materials, (5) understanding

the basic facts linked to the menstrual cycle and (6) how to manage it with dignity and without discomfort or fear.[5] A recent systematic review of menstrual experiences among women and adolescent girls globally highlighted that interventions must address a broad range of issues including addressing stigma, knowledge, social support, restrictive behavioural expectations, and the physical and economic environment.[6] Aligned with this, the term menstrual health and hygiene (MHH) encompasses both MHM and the broader systemic factors that link menstruation with health, well-being, gender equality, education, equity, empowerment and rights.[5]

Effective MHH interventions may lead to sustained benefits for health, productivity[3 4 7] and the environment[8] but there has been little rigorous evaluation of interventions to guide policies.[9–11] Schools provide an important setting for addressing MHH challenges concerning stigma, lack of menstrual literacy and goods (menstrual products, improved water, sanitation and hygiene (WASH) facilities and pain management).[12] In 2014, the 'MHM in Ten' initiative developed an agenda for addressing the barriers facing girls in schools in low-income contexts.[13] The first priority of the initiative is to generate rigorous evidence on whether improving MHM improves school attendance.[9 10 14] A systematic review published in 2016[15] concluded there was promising evidence of the effectiveness of MHH interventions on educational outcomes, but insufficient evidence of effect due to a small number of trials (n=8), a high risk of bias and substantial heterogeneity. An earlier systematic review[7] of intervention and observational studies found that of the 11 studies (only one randomised controlled trial (RCT)) investigating the association between MHM and urogenital infections, seven found an increased risk associated with 'worse' MHM (defined differently for each study but generally meaning not using disposable sanitary pads), one found the reverse to be true (an increased risk from using disposable sanitary pads) and three found no association.

The aim of this study was to pilot test a multicomponent school-based MHH intervention ('MENISCUS') and to prepare for a future cluster-randomised trial (CRT) which will evaluate the impact of the intervention on secondary school attendance, performance, menstruation knowledge, health and well-being outcomes in Uganda. Formative work in Wakiso District, Uganda, showed that secondary school girls reported substantial embarrassment and fear of teasing related to menstruation, reporting that this, together with menstrual pain and lack of effective materials for MHM, led to school absenteeism, especially in schools with low socioeconomic catchment populations.[16] The issue is recognised by the Ugandan Government, which has the political will to improve MHM.[17] However, the Government MHM guidelines for schools are not consistently implemented[18] for reasons including lack of funding, unclear roles and responsibilities for MHM, with issues largely left to the senior teachers.

The objectives of the pilot study are to: (1) assess the feasibility and acceptability of implementing a combined package of MHH intervention elements developed in the formative work, delivered to a whole secondary school year for 9 months; (2) assess the outcomes of the intervention package at baseline and endline (knowledge and attitudes towards puberty, menstruation and pain management); (3) pilot the use of daily diaries to estimate school attendance, and compare attendance with estimates using registers, observation visits and retrospective self-report and (4) estimate school retention over a 9-month period.

## METHODS

### Study setting and participants

This pilot study was conducted in Entebbe Municipality in Wakiso District, Uganda. Entebbe has 13 registered secondary schools. Two day schools (one government and one private), both with students of low socioeconomic status (SES), were purposively selected. In this context, low SES schools are characterised by parents with low income, lower education achievement and under resourced in terms of low grade of teachers and facilities such as infrastructure. In these schools, poor student–teacher relationship and personal hygiene were common, and are key barriers to good MHM practices and academic performance. Eligible participants were all male and female students in the second school year (secondary 2 (S2)). Written informed consent was sought from students aged ≥18 years, and from the parents/caretakers of those aged <18 years, with student assent. We also received written informed consent from teachers and parents/caretakers participating in interviews. Participation was voluntary, and confidentiality was ensured by conducting interviews in a private setting and keeping the data collected on secure servers without identifying information.

### Intervention package

We previously conducted a study in this setting to develop the intervention and found that an effective MHH intervention needs to address stigma, education, attitudes and psychosocial well-being, and provide the goods defined above.[16] Following this, we finalised the pilot MENISCUS intervention, informed by social cognitive theory (SCT) which postulates that learning occurs in a social context and that personal factors, behavioural patterns and environmental aspects have bidirectional influences on one another.[19] Using the core constructs of SCT, we developed a theory of change (ToC) (figure 1) with stakeholders to articulate the aims of the intervention: (1) increase girls' self-efficacy to manage their menstruation (eg, through provision of an MHM kit and pain-management options); (2) use observational learning to reinforce girls' learning and that of boys, teachers and parents to create a more supportive MHM environment (through drama) and (3) provide positive reinforcement for behavioural

## Figure 1: Theory of change for the MENISCUS intervention

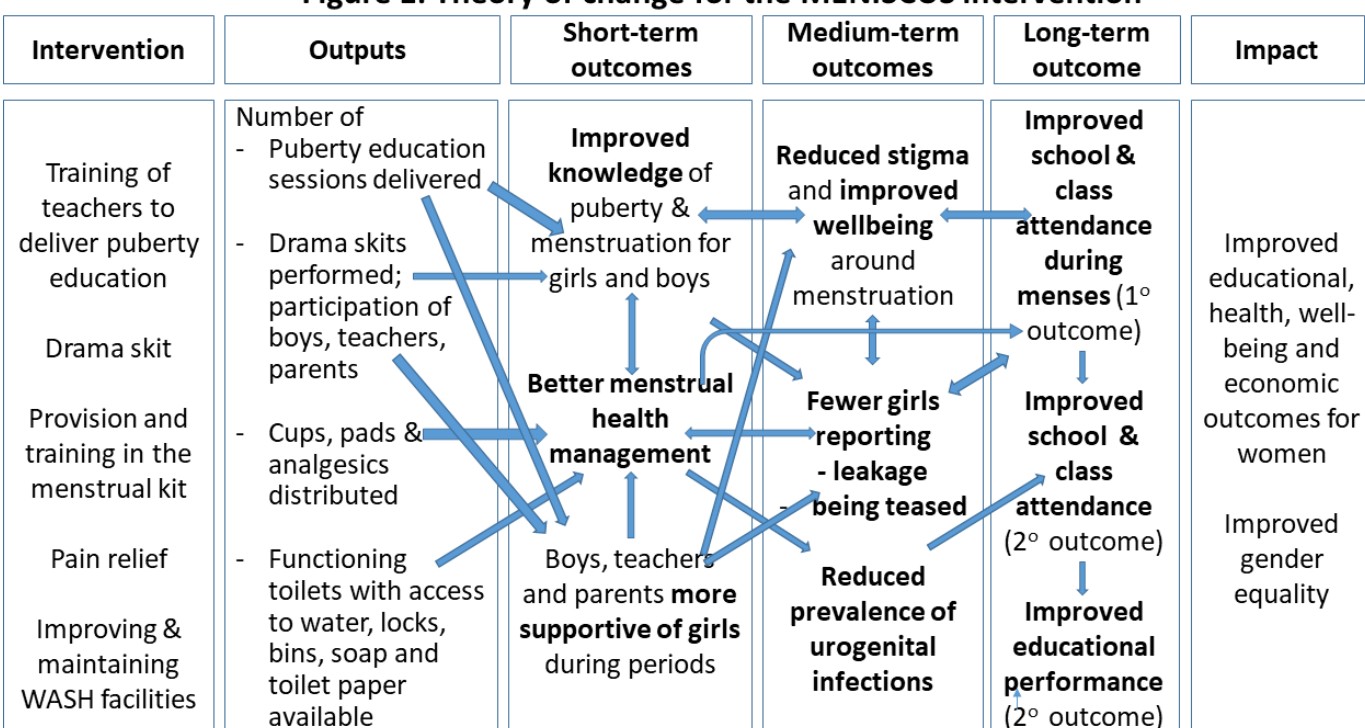

**Figure 1** Theory of change for the meniscus intervention. WASH, water and sanitation hygiene.

change (through improving WASH facilities and reduced leakage/pain as a result of better MHM).

The intervention was implemented in partnership with a local non-governmental organisation (NGO), WoMena Uganda, and consisted of the following. Further details are given in online supplementary table 1:

1. Puberty education: We used a training of trainers model in which WoMena Uganda worked with an educational consultant and her assistant to train 30 teachers from the two schools to improve delivery of Ugandan Government guidelines for puberty education to their students. The trained teachers were expected to develop a plan for delivering puberty training in their school and deliver the specified number of sessions to both male and female students.

2. Drama skit: To address stigma, attitudes and psychosocial issues about menstruation, WoMena Uganda facilitated students, with support from the school drama teacher, to develop and perform drama skits. Topics included managing pain, teasing, parental responsibilities for supporting girls with menstruation and tracking the menstrual cycle.

3. Menstrual management kit: A kit, including a pack of four AFRipads reusable menstrual pads, a small towel, soap, water bottle, underwear, a mirror and menstrual calendar, was distributed to all girls in S2. WoMena Uganda trained teachers and prefects to deliver education sessions on menstruation to girls and boys on MHH, pain relief strategies such as stretching, using a hot-water bottle and use of the reusable pads to girls only.

4. Pain management: The research team supplied each girl with vouchers to exchange for up to six paracetamol tablets per month from the school nurse, senior teacher or a local pharmacy.

5. WASH facilities: The research team improved school WASH facilities by improving access to water close to the sanitation facilities, for example, provision of water drums and stands (one with clean water and one with soapy water), installing locks, repairing broken doors, providing bins and providing toilet-paper holders.

### Study design and evaluation

The design was a longitudinal study with pre–post evaluation of a pilot intervention. All participants were asked to self-complete a quantitative survey at baseline (October 2017) and endline (August 2018) on tablets using Open-Data-Kit software. Questions included knowledge of puberty and menstruation, perceptions of menstruation (using a 5-point Likert scale) (table 1), menstrual management practices at last menstrual period (LMP), reported pain and pain management strategies, leakages during LMP, anxiety and the 25-item Strengths and Difficulties Questionnaire (SDQ) which is globally used to screen for behavioural markers of potential mental health problems in children and adolescents, covering four domains (hyperactivity, conduct problems, peer problems and emotional problems).[20–22]

A nested cohort comprised a random subsample of 50 girls per school assessing the feasibility of school attendance and menstruation patterns using self-completed daily diaries. Nested cohort participants were given a

**Table 1** Menstruation knowledge, myth and perception questions

| Topic | |
|---|---|
| 1. Menstruation knowledge statements (response true/false) | Adolescence is the time between puberty and adulthood. |
| | Changes in the body during puberty happen because of hormones. |
| | The physical changes related to puberty usually start between 10 and 14 years of age in girls, and between 12 and 16 in boys. |
| | Menstrual blood comes from the stomach where food is digested. |
| | Women stop menstruating after the age of about 45–50. |
| | Menstruation in girls and women is normal. |
| | Pregnant women menstruate. |
| | When a girl gets her first period, her body is ready to have children. |
| | During her period a girl can get pregnant. |
| 2. Myth statements (response true/false) | Painkillers cause problems having children. |
| | When a girl has her period she is unclean. |
| | Sanitary pads can cause sickness or infection. |
| | It is healthy for a girl to run, dance or cycle during her period. |
| 3. Menstrual cycle questions (closed responses) | What is period blood? |
| | How long does a period usually last? |
| | How many days are there usually between periods? |
| 4. Menstruation perception statements (response (yes/no)) | I prefer staying at home during my period rather than going to school. |
| | I worry about being teased during my period. |
| | During my period I feel less self-confident than during other days. |
| | During my period I avoid physical activity (eg, walking, running). |
| | I feel anxious about having my next period. |
| | Boys tease me about my period. |
| | Girls tease me about my period. |
| | I feel comfortable to talk to other girls at school about my period. |
| | If I had a problem with managing my period, I would talk to another girl about it. |

booklet each term for daily recording of school and class attendance, menstruation and pain.

Qualitative interviews were used to assess perceptions and acceptability of the intervention, and its perceived impact on school attendance at endline. In-depth interviews (IDIs) were conducted with 20 female students sampled purposively (menstruating S2 girls with different levels of reported baseline school attendance), 10 teachers (6 females and 4 males) and 10 parents who had attended the drama performance. Four focus group discussions (FGDs) were conducted (two with each sex) at the schools. Semistructured topic guides were used for both IDIs and FGDs. Interviewers were the same sex as respondents and interviews were digitally recorded with permission. The trustworthiness of the qualitative data was assessed in accordance with the approaches suggested by Krefting[23] to ensure credibility, applicability, dependability and confirmability of the data. The training, skill and careful supervision and support of researchers in the conduct of qualitative data collection underpins this approach.[24]

Research assistants conducted nine unannounced WASH assessment visits per school at baseline and during follow-up, using a standardised checklist to assess availability, accessibility and functioning in terms of sanitary waste disposal, availability of water, soap, hand-washing facilities and toilet paper; cleanliness and privacy.

Finally, a mixed-methods process evaluation was conducted to improve understanding of intervention implementation. Details of this will be reported separately.

### Outcomes

Outcomes were knowledge of puberty and menstruation, attitudes towards menstruation and menstrual practices (assessed by the proportion of students answering all knowledge, myth and menstrual cycles correct, respectively), knowledge and use of effective pain management methods (assessed by the proportion of girls knowing, and reporting using, at least one effective pain relief method during their LMP if they reported pain), psychosocial well-being (assessed by the mean SDQ-25 score) and school attendance (assessed by reported days missing school in the daily diaries). The planned sample size of

200 girls (and 200 boys, respectively) provides 85% power to detect an OR of two between baseline and endline assuming baseline prevalence of 20%, allowing for within-individual correlation (intracluster correlation=0.05).

### Data management and analysis

Electronically captured data were exported to Stata V.15. The paper-based diary data and WASH checklist data were entered using Microsoft Access. For the survey data, the intervention effect on binary outcomes was estimated by comparing endline versus baseline measures, using adjusted prevalence ratios (APRs), adjusted prevalence difference (APD) and 95% CIs estimated using marginalised standardisation from random-effects logistic regression accounting for within-individual clustering.[25] This provides an estimate of the prevalence ratio and prevalence difference for outcomes assessed at endline compared with baseline, independently of potential confounding variables. Effects on continuous outcomes were similarly assessed with random-effects linear regression to estimate the adjusted mean difference and 95% CI. All analyses were adjusted for school, gender and age (<16, >16 years) as fixed effects. Effect moderation by school and age group were assessed using the likelihood ratio test. For the nested cohort, period days were defined from the diary data as the days of menstruation (bleeding) plus the day prior to menstruation, with sensitivity analyses restricted to days of menstruation only. For the cohort analysis, random-effects logistic regression was used to estimate APR and APD, adjusting for within-girl clustering. Audio recordings were transcribed verbatim and translated into English. Data were analysed using thematic content analysis, with two social scientists independently coding and checking results. Key themes and subthemes related to the objectives of the qualitative research were organised in a matrix and discussed by the team for appropriate interpretation.

### Participant and public involvement

Community groups were involved throughout the formative work, with stakeholders' workshops in August 2016 and October 2018, and a ToC workshop in April 2017. The ToC workshop included 35 invited participants (teachers, students and parents from the MENISCUS-1 and MENISCUS-2 schools, representatives from the Ministry of Education and Sports (MoES), Ministry of Health (MoH), the District Education Officer, Makerere University and NGOs working on menstrual health). In October 2018, we disseminated findings from MENISCUS-2 at a stakeholders' workshop, and elicited input into the future trial design. This meeting was attended by 60 participants, including representatives from MoES, MoH, Uganda National Council for Science and Technology, Wakiso District Local Government Makerere University, AFRIpads, WoMena and Entebbe Municipal Council. MENISCUS-1 and two schools were represented by head teachers, students and parents.

## RESULTS

### Sociodemographic characteristics

Of the 473 eligible S2 students at baseline, 450 (95.1%; 232 girls and 218 boys) consented/assented. Overall, 369 students (188 girls; 81.0%; and 181 boys; 83.0%) participated in the endline survey. At baseline, the mean age was 15.4 years (SD 1.31; range 12–20) for girls and 16.2 (SD=1.5; range 13–21) for boys. Follow-up ranged from 8.6 to 9.6 months (median 9.00 months). Among the girls, 222 (95.7%) and 183 (97.3%) had started menstruating at baseline and endline, respectively. The majority (n=361; 80.2%) were Christian and 198 (44.0%) were of Ganda ethnicity. About half (50.6%) of participants' mothers had postprimary education, as did 73.5% of participants' fathers (excluding 74 mothers and 88 fathers where this information was unknown). Relatively few participants (n=59; 13.1%) had running water at home, and 54 (12.0%) had a flush toilet in the home.

### Feasibility and acceptability of implementing the intervention

There were high levels of uptake of the individual and behavioural intervention components (puberty education, drama skit, MHM kit and pain management). Most students (81.3%) reported attending puberty education sessions and 93.4% of girls reported receiving the MHM kit. Both schools performed the drama skit, and the qualitative interviews indicated that the skit increased MHM awareness and enabled some of the girls to talk about MHM with their parents, especially their fathers.

> When we all saw the skit, it gave us a starting point to initiate a discussion with our children. Even men got it (Female Parent).

> Trainings improved our self-esteem and confidence because nowadays we are not scared of coming to school. We are comfortable coming to school during our menstruation. Before the training some of us were shy and we couldn't stand and talk in-front of people or parents about menstruation and puberty but now we can (FGD girls, Private school).

The WASH component was challenging to implement. At baseline, there were no sanitary disposal bins, lockable toilet paper holders or functional water drums at either school (table 2). Both schools received these items as part of the intervention. At endline, in the private school, all the girls' cubicles were functional, clean and with a lockable door versus only 33% in the government school, where facilities were shared with the community. Despite this, most girls (81.4%) reported improvements in toilet facilities at endline, which improved their comfort managing menstruation.

> The school environment is now conducive … during our menstruation because the water is available, they give us water mixed with liquid soap; the toilet doors have locks, there is privacy, a person cannot interrupt you while changing the pad. Toilet paper is available

**Table 2** Summary of wash facilities at baseline and during implementation

| | Baseline (one visit) | Follow-up (eight visits) | | | |
| | Boys and girls | Girls | | Boys | |
| Component | Both schools | Private school | Government school | Private school | Government school |
|---|---|---|---|---|---|
| Bin | 0% | 6/8 (75%) | 8/8 (100%) | n/a | n/a |
| Toilet paper | 0% | 2/8 (25%) | 0/8 (0%) | 1/8 (13%) | 1/8 (13%) |
| Functioning water drum | 0% | 7/8 (88%) | 7/8 (88%) | 7/8 (88%) | 7/8 (88%) |
| Functioning water and soap drum | 0% | 6/8 (75%) | 2/8 (25%) | 5/8 (63%) | 3/8 (38%) |

Per cent of the eight visits when the WASH component was present outside at least one toilet block in the school.
n/a, not applicable.

whenever needed and the toilets are always clean (Female student, Private school).

### Knowledge, misconceptions and perceptions of puberty and menstruation

Knowledge of puberty and menstruation was poor at baseline, especially among boys. The proportion answering all nine knowledge questions correctly increased from 11.6% to 23.9% (APR=2.18, 95% CI 1.47 to 3.22; p<0.001) in girls, and from 4.1% to 12.7% (APR=3.07, 95% CI 1.49 to 6.32; p<0.001) in boys (table 3). There was also evidence of improvements in knowledge of the menstrual cycle, particularly among girls, although endline knowledge remained poor (29.3% of girls and 7.7% of boys answering all three questions correctly; table 3).

The proportion of girls answering at least eight of nine questions on their perceptions about menstruation positively (ie, questions about self-confidence, teasing and anxiety) increased from 12.2% at baseline to 27.9% at endline (APR=2.40, 95% CI 1.59 to 3.62, p<0.001). The largest effects were reductions in the proportion of girls reporting teasing by boys about menstruation (14.4%–8.7%, APR=0.57, 95% CI 0.34 to 0.97, p=0.04), girls feeling anxious about the next period (58.6%–34.4%, APR=0.57, 95% CI 0.46 to 0.69, p<0.001) and girls avoiding physical activity during menstruation (47.8%–25.7%; APR=0.55,

**Table 3** Knowledge of puberty and menstruation, attitudes to menstruation and menstrual practices at baseline and endline, by gender

| | Girls | | | | | |
| | Baseline (n=232) | Endline (n=188) | APR (95% CI) APD (95% CI) | Baseline (n=218) | Endline (n=181) | APR (95% CI) APD (95% CI) |
|---|---|---|---|---|---|---|
| All nine knowledge questions correct* | 27 (11.6%) | 45 (23.9%) | 2.18 (1.47 to 3.22) 13.4% (6.7% to 20.2%) p<0.001 | 9 (4.1%) | 23 (12.7%) | 3.07 (1.49 to 6.32) 8.5% (3.1% to 13.8%) p=0.002 |
| Eight knowledge questions (excluding fertility question) correct* | 70 (30.2%) | 120 (63.8%) | 2.13 (1.73 to 2.63) 34.1% (25.7% to 42.0%) p<0.001 | 42 (19.3%) | 73 (40.3%) | 1.91 (1.44 to 2.54) 18.3% (10.7% to 26.0%) p<0.001 |
| All four myth questions correct† | 39 (16.8%) | 77 (41.0%) | 2.54 (1.86 to 3.48) 25.3% (17.3% to 33.3%) p<0.001 | 22 (10.1%) | 44 (24.3%) | 2.56 (1.64 to 4.01) 15.2% (8.1% to 22.2%) p<0.001 |
| All three menstrual cycle questions‡ | 12 (5.2%) | 55 (29.3%) | 6.38 (3.59 to 11.34) 25.8% (18.8% to 32.8%) p<0.001 | 4 (1.8%) | 14 (7.7%) | 3.70 (1.23 to 11.11) 5.3% (1.2% to 9.4%) p=0.02 |

*Knowledge statements were: (1) Adolescence is the time between puberty and adulthood; (2) Changes in the body during puberty happen because of hormones; (3) The physical changes related to puberty usually start between 10 and 14 years of age in girls, and between 12 and 16 in boys; (4) Menstrual blood comes from the stomach where food is digested; (5) Women stop menstruating after the age of about 45–50; (6) Menstruation in girls and women is normal; (7) Pregnant women menstruate; (8) When a girl gets her first period, her body is ready to have children; (9) During her period a girl can get pregnant.
†Myth statements were: (1) Painkillers cause problems having children; (2) When a girl has her period she is unclean; (3) Sanitary pads can cause sickness or infection; (4) It is healthy for a girl to run, dance or cycle during her period.
‡Menstrual cycle questions were: (1) What is period blood?; (2) How long does a period usually last?; (3) How many days are there usually between periods?.
APD, adjusted prevalence difference; APR, adjusted prevalence ratio.

**Table 4** Reported perceptions of menstruation, menstrual management, pain and pain management among girls, by age at baseline

| | Age <16 years | | | Age ≥16 years | | |
|---|---|---|---|---|---|---|
| | Baseline (n=115) | Endline (n=51) | APR (95% CI) APD (95% CI) | Baseline (n=107) | Endline (n=132) | APR (95% CI) APD (95% CI) |
| ≥8 of 9 positive perceptions of menstruation* | 14 (12.2%) | 16 (31.4%) | 2.76 (1.10 to 25.19) 21.0% (7.2% to 34.9%) p=0.001 | 13 (12.2%) | 35 (26.5%) | 2.23 (1.26 to 3.95) 14.6% (5.2% to 24.1%) p<0.001 |
| Used manufactured menstrual materials only at LMP | 86 (74.8%) | 46 (90.2%) | 1.21 (1.06 to 1.39) 15.7% (4.6% to 26.8%) p=0.006 | 76 (71.0%) | 117 (88.6%) | 1.25 (1.10 to 1.42) 17.6% (8.0% to 27.2%) p<0.001 |
| Leaked blood at LMP | 37 (50.0%) | 12 (42.9%) | 0.84 (0.54 to 1.30) −8.2% (−27.5% to 11.1%) p=0.41 | 37 (45.1%) | 24 (32.4%) | 0.73 (0.49 to 1.09) −12.1% (−27.1% to 2.8%) p=0.11 |
| Underwear stained at LMP | 31 (27.0%) | 16 (31.4%) | 1.07 (0.67 to 1.71) 18.4% (−11.5% to 15.1%) p=0.79 | 21 (19.6%) | 34 (25.8%) | 1.28 (0.82 to 1.99) 5.6% (−4.3% to 15.5%) p=0.27 |
| Knew ≥4 effective pain management methods | 30 (24.2%) | 37 (67.3%) | 2.72 (1.93 to 3.84) 41.8% (27.9% to 55.7%) p<0.001 | 30 (27.8%) | 97 (72.9%) | 2.60 (1.93 to 3.51) 44.5% (34.3% to 54.7%) p<0.001 |
| Reported pain at last period | 79 (68.7%) | 36 (70.6%) | 1.00 (0.82 to 1.22) −0.01% (13.6% to 13.4%) p=0.99 | 86 (80.4%) | 94 (71.2%) | 0.89 (0.78 to 1.01) −9.1% (−19.1 to 0.9%) p=0.08 |
| Used ≥1 effective pain management method† | 57 (72.2%) | 34 (94.4%) | 1.31 (1.12 to 1.54) 22.5% (10.2% to 34.9%) p<0.001 | 69 (80.2%) | 85 (90.4%) | 1.13 (1.00 to 1.27) 10.0% (2.9% to 19.9%) p=0.05 |
| Used painkillers at LMP‡ | 32 (40.5%) | 21 (58.3%) | 1.45 (0.99 to 2.12) 18.2% (−1.6% to 37.5%) p=0.06 | 45 (52.3%) | 58 (61.7%) | 1.16 (0.92 to 1.46) 8.3% (−4.5% to 21.1%) p=0.21 |
| Used other effective methods‡ | 39 (49.4%) | 29 (80.6%) | 1.63 (1.24 to 2.15) 31.1% (14.0% to 48.1%) p<0.001 | 43 (50.0%) | 72 (76.6%) | 1.55 (1.25 to 1.92) 27.3% (15.2% to 39.4%) p<0.001 |
| Did nothing for pain at LMP† | 20 (25.3%) | 2 (5.6%) | 0.22 (0.05 to 0.89) −19.7% (−31.9% to −7.5%) p<0.001 | 17 (19.8%) | 7 (7.5%) | 0.36 (0.17 to 0.78) −12.8% (−21.8% to 3.7%) p<0.001 |

*Perception questions were (1) I prefer staying at home during my period rather than going to school; (2) I worry about being teased during my period; (3) During my period I feel less self-confident than during other days; (4) During my period I avoid physical activity (eg, walking, running); (5) I feel anxious about having my next period; (6) Boys tease me about my period; (7) Girls tease me about my period; (8) I feel comfortable to talk to other girls at school about my period; (9) If I had a problem with managing my period, I would talk to another girl about it.
†Effective methods (painkiller, drinking water, using hot water bottle, exercise, relaxing, foods with lots of water).
‡Among those with pain at LMP.
APD, adjusted prevalence difference; APR, adjusted prevalence ratio; LMP, last menstrual period.

95% CI 0.42 to 0.71, p<0.001). There was no evidence that the intervention effect differed by age (table 4; p≥0.2 for intervention effect modification for each variable) or school (p≥0.1 for effect modification for each variable; results not shown).

In qualitative interviews, girls and teachers said that they thought that fear, myths and negative perceptions about menstruation were due to lack of knowledge. Participants suggested that the intervention improved knowledge about puberty, menstruation, and girls' accounts suggested a reduction of teasing by boys at school:

Before MENISCUS, boys used to laugh at girls, for example when a girl stood up in class with her dress stained, boys would laugh at her but after MENISCUS training, they stopped laughing at girls and they now care about us (Female student, Private school).

**Management of menstruation**

The proportion of girls reporting using manufactured menstrual materials exclusively (ie, reusable or disposable pads, tampons or menstrual cups) during their LMP increased from 73.0% at baseline to 89.1% at endline (APR=1.23, 95% CI 1.12 to 1.35, p<0.001). At endline, most girls (n=155; 82.5%) reported using reusable pads during their LMP compared with 18.5% at baseline. There was weak evidence of a decrease in reported leakage of blood through their clothing during LMP (47.4% at baseline vs 35.3% at endline; APR=0.76, 95% CI 0.57 to 1.02, p=0.06), but no evidence of a difference in the proportion of girls reporting staining their underwear during their LMP (23.4% at baseline to 27.3% at endline (APR=1.19, 95% CI 0.87 to 1.63, p=0.29). There was no evidence of intervention effect moderation by age or school (table 4).

### Knowledge and use of effective pain management methods

Most girls reported pain during the menstruation at both baseline and endline (74.3% and 71.0%, respectively). Among these girls, there was evidence of an increase in the proportion who reported using painkillers during their LMP (46.7% at baseline to 60.8% at endline; APR=1.26, 95% CI 1.03 to 1.55, p=0.03). Results were similar by age (table 4) and school (results not shown). There was evidence of an increase in the proportion of girls who reported using ≥1 effective pain relief method during their LMP (76.4% at baseline to 91.5% at endline; APR=1.19, 95% CI 1.08 to 1.32, p=0.001). Of the 232 girls who received a voucher for painkillers, 58 (25.0%) had redeemed 78 vouchers by endline: 77% used one voucher, 16% used two and 7% used ≥2.

The qualitative findings confirmed that non-pharmacological methods of pain relief were popular and effective.

> They [MENISCUS team] taught us how to do exercises to relieve pain and it worked for me so, the last time, I didn't use painkillers; I managed my periods by doing exercises and using reusable pads MENISCUS had provided. (Female student, Government school).

### Behavioural markers of psychosocial well-being

The mean SDQ-25 score in girls decreased from 10.3 to 9.2 (p=0.006, adjusted for age and school), indicating improved behaviour and conduct. No decrease was seen among boys, in line with the ToC (9.88 at baseline vs 9.91 at endline; p=0.98).

### Menstruation and school absence

In the cross-sectional surveys, a similar proportion of girls reported missing at least one school day due to menstruation in Term 2, 2017 (prebaseline) and term 2, 2018 (endline) (32.0% vs 32.8%; APR=0.99, 95% CI 0.76 to 1.28, p=0.92). Among the 100 cohort participants, data were collected from 81 students at endline. The APR associated with missing school on period-days compared with non-period days decreased from 1.84 (95% CI 1.46 to 2.21) at baseline to 1.16 (95% CI 0.97 to 1.38) at endline (p value for moderation=0.01; table 5). Results were similar for class attendance (table 5). The direct observation checks found high correlation with the self-completed diaries. Girls were seen on 328/330 (99.4%) of days when their diary stated they were present and were not seen on all 37 days when the diary stated that they were absent. School registers were incomplete and rarely completed by teachers.

The qualitative interviews suggested that girls were more likely to attend school during menstruation at endline than at baseline, with reasons including the training on pain management, tracking their menstrual cycle, and having reusable pads.

> Some girls totally don't come to school every menstruation period. However, in our class, missing school is now rare because we have learnt how to manage the periods, pain and we have what to use. We now also know our cycle (Girls FGD; Government school).

> Before MENISCUS study gave us diaries, I didn't know how to track my days. I used to come to school and my uniform would get soiled. Sometimes, I could ask for a pass out to go back home during my periods. […] But when MENISCUS came, they gave us diaries, I started filling them in and now I know my days; I come to school prepared. (Female, Private school).

## DISCUSSION

To our knowledge, this is the first MHM intervention to be pilot tested that focused equally on health promotion (psychosocial issues, stigma and knowledge) and goods

**Table 5** School and class attendance at baseline and endline among nested cohort participants, by period day status

| No (N) | Term 3 2017 (baseline) 99 girls | | Term 2 2018 (endline) 81 girls | |
|---|---|---|---|---|
| | Non period day | Period day* | Non period day | Period day |
| No of days | 2625 | 554 | 4111 | 838 |
| N (%) not attending full day of school | 8.5 | 14.6 | 12.7 | 14.8 |
| APR (95% CI)† APD (95% CI)† | 1.84 (1.46 to 2.31) 7.1% (3.1% to 10.2%) p<0.001 | | 1.16 (0.97 to 1.38) 2.1% (−0.5% to 4.6%) p=0.10 | |
| N (%) not attending all classes | 11.4 | 19.9 | 18.2 | 20.6 |
| APR (95% CI)† APD (95% CI)† | 1.79 (1.47 to 2.17) 8.9% (5.4% to 12.5%) p<0.001 | | 1.15 (0.99 to 1.32) 2.7% (−0.3 to 5.6%) p=0.07 | |

*Includes days of menses plus day prior to menses (results similar when restricted to days of menses) obtained from daily diaries for 9 months.
†Obtained from random-effects logistic regression adjusting for age and school, allowing for within-girl clustering.
APD, adjusted prevalence difference; APR, adjusted prevalence ratio.

(provision of pads and analgesics, and improved WASH facilities).[26]

### Feasibility and accessibility of implementing the intervention

The qualitative and quantitative data showed that the intervention was feasible and acceptable to schools and stakeholders. A detailed process evaluation will be published separately. The main challenges with implementation was maintenance of the WASH component and that we did not obtain ethical approval to offer girls a menstrual cup, due to concerns about whether cup insertion could lead to damage of the hymen and affect a girl's virginity.

### Potential impact of an MHH intervention on knowledge and attitudes to puberty and menstruation

The study highlighted poor baseline knowledge of menstruation, possibly due to cultural taboos and norms,[16 27] and lack of knowledge among students[28] parents and teachers hindering discussion of puberty and menstruation.[16 29–31] Three intervention components addressed knowledge (puberty education, menstrual-management kit training and drama skit). To address stigma and effect a school-wide change in attitudes, the education and drama sessions included boys. The qualitative results indicated a preference for a boys-only puberty education session. This pilot study showed that girls were less anxious about menstruation, and reported less teasing by boys after the intervention, supporting the ToC and previous qualitative work from Ghana.[32] It also worth noting that although the SDQ has proven reliability and validity in a number of studies across Europe, Asia, Australia and South America,[21 33 34] and has been used widely in Africa,[22] there has only been one psychometric validation of the tool which found satisfactory internal consistency.[35]

At endline, compared with baseline, more girls reported use of only manufactured materials during their LMP, and using these correctly. The high uptake of reusable pads was expected, as these were provided free to all girls, but the continued use over follow-up, and favourable qualitative reports, shows that pads were acceptable. Similar findings on correct use have been seen in another study in Uganda where 98% of girls reported washing and using soap for their AFRIpads[26] and in India,[36] where a school-based health-education intervention led to improvements in washing clothes with soap, drying them in the sun and safe disposal.

Overall, the findings suggest a potential intervention effect on education and health outcomes, but the results need to be interpreted cautiously as the improvements reported may be partly due to (1) older age at endline—there may have been improvements in knowledge and self-confidence in managing menstruation due to a cohort effect; (2) reporting biases—the outcome assessment team were involved with implementation and this may have caused social desirability bias affecting responses on attitudes or behaviours following the intervention; (3) biases due to the lack of a comparison group, and

the endline survey being conducted in a different term to baseline. For example, girls reported higher levels of school attendance overall at baseline than at endline, and this may be due to the baseline being conducted during an examination term and (4) generalisability—the extent to which are findings will hold in other geographic or socioeconomic settings is not clear.

### Methods to estimate school attendance

We found strong evidence that the association between missing school and menstruation was smaller by the end of the intervention period. This aligns with data from qualitative interviews, in which girls attributed the improved school attendance during menstruation to improved pain management, knowledge to track menstrual cycles and the provision of reusable pads. Previous studies show inconsistent evidence for a relationship between menstruation and school absenteeism.[15 16 37–44] Documenting school attendance is often challenging in LMICs due to incomplete or inaccurate school registers.[7 16] Our diaries, administered by the research team, were popular and correlated very well with direct observation.

### Key recommendations

#### Multicomponent focus

Past and ongoing studies tend to focus on single-component interventions.[7 40] This study indicates synergies between the intervention elements, for example, the drama skit reinforces puberty knowledge, reduces stigma and engages parents and boys. An emerging view, aligned with our findings, is that poor MHH is a social problem and to be effective, MHH interventions in many settings should address the broader issues of menstrual stigma and literacy as well as the provision of menstrual products or improving WASH facilities.[9] A key finding from this study is that the multicomponent approach addresses both individual, behavioural and environmental barriers to good MHH and school attendance. Based on our findings, these holistic MHH interventions may benefit from establishing an MHM leadership group (to include teachers, parents and students) responsible for ensuring each intervention element is delivered, maintained and sustained by the schools. We recommend that future MHH interventions are multicomponent, which is also in line with a recent systematic review of girls' and women's experiences of menstruation in LMICs.[6] Given the limitations of this study mentioned above, a full-scale CRT is needed to rigorously evaluate the impact of such an intervention on girls' health and well-being.

#### Inclusion of pain management strategies

Menstrual pain is a major contributor to school absence in girls in many settings.[16 27 45 46] This is the first MHH intervention to address this issue and we found substantial improvements in pain management (using both analgesic and non-analgesic methods).[16] Further research is needed on this topic as misconceptions about use of painkillers are common in Uganda[16] and elsewhere.[47 48]

## Inclusion of boys

We recommend that MHH interventions are inclusive of boys and men. Most MHH interventions focus only on girls, yet sustainable changes in MHM depend on addressing stigma about menstruation. This study highlighted the importance of including boys which is aligned with SCT (ie, the intervention provides positive reinforcement for behavioural change by improving the school environment). The focus on boys as well as girls is supported by WHO framework for Health Promoting Schools[49] and is aligned with a report commissioned by the Gates Foundation on Menstrual Health and Gender Equity.[50]

## Implementation in diverse settings

Most intervention studies to date have been in primary schools and in rural areas.[26 45] MHH and school attendance is an issue in the periurban area in Entebbe where most girls can afford to buy disposable pads for some of the time.[16] A focus on MHH in secondary schools in different settings is important given the recognised importance of girls' secondary education to future development,[51] and evidence that school absenteeism due to menstruation is problematic in secondary school girls.[16 52]

## Piloting and validation of methods to assess the association between menstruation and school attendance

In this study, daily diaries were the optimal method to assess the association of school attendance and menstruation. This may also be the case in other settings where school registers are not reliable, although they place some burden on the girls, and may not be accurately filled if there is little trust with the people collecting the data. Diaries have the advantage of providing data on menstruation for each girl, to enable the link with school attendance, and in this study, girls used the diaries to track their periods which reduced anxiety.

## CONCLUSION

This pilot study showed that the MHH intervention was acceptable and feasible to implement. There were substantial reported improvements in stigma and anxiety, pain management, and some improvement in MHH knowledge of puberty and menstruation and WASH following a multicomponent intervention. Based on SCT, we anticipate the results to be generalisable to other similar contexts with relatively low levels of MHM knowledge, stigma about discussing menstruation and poor school WASH facilities. A phase-III trial is warranted to evaluate the impact of the intervention on school attendance, and on health, well-being and educational outcomes for definitive results to drive forward policy changes.

**Author affiliations**
[1]Research Unit, Medical Research Council/Uganda Virus Research Institute & London School of Hygiene & Tropical Medicine Uganda, Entebbe, Uganda
[2]WoMena Uganda, Kampala, Uganda
[3]College of Humanities and Social Science, Makerere University, Kampala, Uganda
[4]Epidemiology and Population Health, London School of Hygiene and Tropical Medicine, London, UK
[5]Palm Tree Academy Uganda, Kampala, Uganda
[6]Uganda Virus Research Institute, Entebbe, Uganda
[7]Faculty of Infectious and Tropical Diseases, London School of Hygiene and Tropical Medicine, London, UK
[8]MRC Tropical Epidemiology Group, Faculty of Epidemiology and Population Health, London School of Hygiene and Tropical Medicine, London, UK
[9]Department of Maternal, Newborn, Child and Adolescent Health, World Health Organization, Geneve, Switzerland
[10]Faculty of Public Health and Policy, London School of Hygiene & Tropical Medicine, London, UK

**Acknowledgements** The authors would like to thank the study participants, contact teachers in the two schools, and the Ministry of Education and Sports and Ministry of Health Uganda for their support.

**Contributors** HAW, LH, BT, DAR, JS, CB, SNe, SCF and CA contributed to the overall conception and design of the study. CK, LH, SNa, RN, KN, PN, SNM undertook the data collection. CK and HAW wrote the first draft of the manuscript. All authors contributed to the interpretation of results and drafting of the manuscript. HAW and CT did the statistical analyses. All authors took responsibility for the integrity of the data and the accuracy of the data analysis. All authors read and approved the final manuscript. HAW is the guarantor.

**Funding** This work was supported by a UK Department for International Development (DFID), the National Institute for Health Research (NIHR), UK Medical Research Council (MRC) and Wellcome Trust Joint Global Health Trials Development Grant (number MR/P020283/1).

**Competing interests** None declared.

**Patient consent for publication** Not required.

**Provenance and peer review** Not commissioned; externally peer reviewed.

**Data availability statement** Data are available on reasonable request. Data will be made available in the LSHTM Data Compass repository on request from the corresponding author (Helen Weiss https://orcid.org/0000-0003-3547-7936) from the website https://datacompass.lshtm.ac.uk/.

**ORCID iDs**
Christopher Bonell http://orcid.org/0000-0002-6253-6498
Helen Anne Weiss http://orcid.org/0000-0003-3547-7936

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
