## [Reviewer comments · BMJ Open]

ARTICLE DETAILS

TITLE (PROVISIONAL)	Menstrual health intervention and school attendance in Uganda (MENISCUS-2): A pilot intervention study
AUTHORS	Kansiime, Catherine; Hytti, Laura; Nalugya, Ruth; Nakuya, Kevin; Namirembe, Prossy; Nakalema, Shamirah; Neema, Stella; Tanton, Clare; Alezuyo, Connie; Namuli Musoke, Saidat; Torondel, Belen; Francis, Suzanna C; Ross, David; Bonell, Christopher; Seeley, Janet; Weiss, Helen Anne

VERSION 1 – REVIEW

REVIEWER	Amal K Halder Oxfam GB, South Sudan
REVIEW RETURNED	02-Jun-2019

GENERAL COMMENTS	A. Abstract -  1. Line # 45-48: “deliver from the qualitative and quantitative data” is a broad term, needs to be specific. 2. The result section in the abstract is missing the quantitative data. 3. Conclusion (line # 5-6) to be specific of what “potentially effective on education and health outcomes”? B. Objectives (page 5, line # 36-42) -  1. Objectives have many outcomes in which some are vague for example the study is missing specific data on impact of education (what education – MHM education or academic subjects?) and wellbeing outcomes (what is the well being outcome?). C. Important limitation is not mentioned – one of the serious limitation is the absence of a control group. D. Data analysis –  1. Covariates and outcome variables needed to be specified. 2. Using adjusted odds ratios --- what was adjusted specifically and why was adjusted? 3. What does adjusted odds ratios mean? How non-statistical audience will understand the value of AOR? 4. How do the authors interpret 95% CI for non-statistical audience? 5. Similarly for “adjusted mean difference (AMD) and 95% CI”? 6. “All analyses were adjusted for school, gender and age (<16, >16 years) as fixed effects” need justification of why were for school, gender and age? Then why fixed effects? Why not mixed effects?
---

	7. How the likelihood ratio test would explain for the non-statistical audience? 8. What do they mean by nested cohort? If it is nested cohort then nested correlation effects need to be adjusted using different model for example nested correlation structure etc.  Page 7, line # 42-44: what is 25-item Strengths and Difficulties Questionnaire? This need to explain. E. Result  Comparative data of socio-economic characteristics, feasibility and acceptability of implementing the intervention, and WASH between baseline and endline needed to be presented in a separate table. Knowledge questions, myth questions and menstrual cycle questions in table 1 needed describe in the method section. Page 11, line 3-5: "Knowledge of puberty and menstruation was poor at baseline and endline, especially among boys" How do the authors say this is poor at endline? What do the AOR data means--- needed to clarify in the text? The data in Table 1, do the authors find that the changes between baseline and endline were statistically significant for both boys and girls? Needed to talk in the result section. What are the specific differences between the boys and girls not clearly described in the text in page 11 between line numbers 3-14. Similarly line number 19-35 in page 11, the difference between two age group is not clearly specified from the table 2 data. In table 2, the perception questions and effective methods needed to bring in method section. No quantitative outcome data in regards to objectives (education, health and wellbeing) are presented except school attendance as shown in Table 3. F. Finally --- Since this is a before and after analysis, the authors needed to bring analysis that shows the associated factors (Puberty education, drama skirt, menstrual management kit, pain management, and WASH kits) related to interventions influenced school attendance, health and well being outcomes. i.e. authors needed to do multivariate analysis by adjusting school, gender, age etc. The authors may take the hints of analysis (Table 4) from the paper "Alam M-U, et al. BMJ Open 2017;7:e015508. doi:10.1136/bmjopen-2016-015508"
--	---

REVIEWER	Jing Jing Su Chinese University of Hong Kong
REVIEW RETURNED	13-Jun-2019

GENERAL COMMENTS	Thanks for inviting me to review this manuscript. This is an important health topic. Involving boys, parents, and public is crucial in improving the management of menstruation as well as minimizing the affect of poor management. I hope the comments can help to improve this work. 1. Introduction
--

	-Considering be more specific in stating the aim of the pilot study. To pilot test a multi-component school based MHM intervention and to prepare for a future cluster-RCT is fine. But also be clear with the aim for pilot itself. The results presented flexibility acceptability and effectiveness. The writing of the paper and presentation of those results needs to be guided by study aim. -In the introduction, the authors stated the barriers for menstrual health for low-income context including products, water, facilities. But according to literature, menstrual health management barriers not only limit to products, water, facilities. Many studies also show the perceptions, stigma, lack of knowledge extra also influence menstrual health management. Especially this study mentioned those factor in the methods and results part, therefore, a clear introduction with literature support is recommended. -managing menstruation is largely influenced by cultural and contextual background. Please give some background information to help the reader understand this population and the challenges they are facing. A better understanding of their current practice and availability of menstrual health management program is needed. -This study can also benefit from reviewing of the existing menstrual health programs. Actually there are a number of quasi-experimental intervention control studies published in this area with effective outcomes. Methods -The manuscript stated that students with low social-economic status were purposively selected. 1)What the definition of low social-economic status? 2) Why limit to low-social-economic status, because everyone need menstrual health information especially during their puberty. Ethical consideration wise, how this study would influence the accessibility of menstrual management of students with better social-economic status. 3) How this would affect generalizability of this study? -Intervention. (1) It is stated that "MHM intervention needs to address stigma, education, attitudes, and psychosocial well-being". Again, a better understanding of the problems from the introduction session would better support the content of intervention. (2) Any measures taken to ensure fidelity as one person trained 30 teacher. (3) The intervention talked about one person trained 30 teachers while the results measured students. There are parts missing. I presume the teachers would also teach the students. The study procedure is unclear. The dosage and format of training is unclear. Are boys and girls attending class together? If that's the case, how students take it as you mentioned stigma earlier on. You had content of parental responsibilities for supporting girls with menstruation and also collected data from parent. Did parents also participated in the study? How comfortable the girls are with presence of boys and parents? -Study design and evaluation. No report characteristics especially psychometric property of the instruments used. (1) selection of suitable instruments is crucial for this topic. Questionnaire used in one culture/country may not make sense for people in different culture. Questionnaire used for adults may not be suitable for adolescent as their menstruation characteristics differs. (2) A clear operational definition is also recommended as to help reader understand what the instruments trying to measure. -power analysis is needed in determine the sample size.
--	--

	-who, when, how the qualitative interview is conducted? Any ethical consideration for interview adolescents. -Ethical consideration. Students and adolescents belong to vulnerable group which requires more careful ethical consideration. Please add one session taking about ethical issue. -Patient and public consideration. There are no patients for participation of this study. What is the role of this part in your intervention? Results - Feasibility and acceptability of implementing the intervention. (1) Is this intervention a mandatory course or students choice for free? I am not sure how this is organised, therefore, the feasibility is unclear. (2) The first paragraph talked about how this program enabled some girls to talk about menstruation to their parents especially father while the quote is from the parents side saying they started talking to their child. -Not sure how appropriate the boys can answer some of the questionnaire as the definition and property of questionnaire is missing. Seems SDQ measures behavior. -How the qualitative and quantitative data converged. I mean for example, knowledge improvement in terms of increasing in knowledge questionnaire score can be reflected by their statement of improved understanding of menstruation. -A clear statement of study objective may help structuring the results to improve clarity. -Again, there are many intervention studies with control group using menstrual health programs to improve menstrual health as well as public health initiatives to improve hygienic facilities. BMJ open is an international study, consider the implications this study has for global readers. What are the crucial elements identified as to move forward the menstrual health management programs. Involving boys and parents can be one, but it may need very clear rational and description to make their role explicit.
--	--

VERSION 1 – AUTHOR RESPONSE

Reviewer 1: Amal Halder, Oxfam South Sudan

A. Abstract -

1. Line # 45-48: “deliver from the qualitative and quantitative data” is a broad term, needs to be specific.
2. The result section in the abstract is missing the quantitative data.
3. Conclusion (line # 5-6) to be specific of what “potentially effective on education and health outcomes”?

Response: We have added the key quantitative results to the abstract, which addresses points 1 and 2. We have amended the conclusion to read “..potentially effective in improving menstruation knowledge and management”.

B. Objectives (page 5, line # 36-42) -

1. Objectives have many outcomes in which some are vague for example the study is missing specific data on impact of education (what education – MHM education or academic subjects?) and wellbeing outcomes (what is the wellbeing outcome?).

Response: We have added the specific objectives of the pilot study (Page 5; final paragraph).

C. Important limitation is not mentioned – one of the serious limitation is the absence of a control

group.

Response: We agree that this an important limitation and had included this in the discussion (page 19). We have also added this in the strength and limitation section on page 3 of revised manuscript.

D. Data analysis –

1. Covariates and outcome variables needed to be specified.

Response: We have added a paragraph specifying the outcomes on pages 9-10. The main covariate is time (baseline vs endline) with adjustment as specified (for school, gender and age).

2. Using adjusted odds ratios --- what was adjusted specifically and why was adjusted?

Response: As stated on page 10, we adjusted for school, gender and age (<16, >16 years) in order to adjust for potential confounding of these variables on the outcomes.

3. What does adjusted odds ratios mean? How non-statistical audience will understand the value of AOR?

Response: We have included a sentence to explain the adjusted odds ratio on page 10.

4. How do the authors interpret 95% CI for non-statistical audience?

Response: 95% CI are standard but we can add an interpretation if the editors require this.

5. Similarly for “adjusted mean difference (AMD) and 95% CI”?

Response: This is standard terminology but can explain if required by the editors.

6. “All analyses were adjusted for school, gender and age (<16, >16 years) as fixed effects” need justification of why were for school, gender and age? Then why fixed effects? Why not mixed effects?

Response: We expected that the intervention effect might vary with school, gender and age. There is no clustering, and these are each binary covariates, so fixed effects are more appropriate than random effects.

7. How the likelihood ratio test would explain for the non-statistical audience?

Response: This is standard methodology which is not usually explained within a paper, but we can add if requested by the editors.

8. What do they mean by nested cohort? If it is nested cohort then nested correlation effects need to be adjusted using different model for example nested correlation structure etc.

Response: The nested cohort are the 100 randomly-selected participants who were given a daily diary, as explained on page 9. The random effects model allows for within-individual correlation.

9. Page 7, line # 42-44: what is 25-item Strengths and Difficulties Questionnaire? This need to explain.

Response: We have added a brief explanation on page 8.

E. Result

1. Comparative data of socio-economic characteristics, feasibility and acceptability of implementing the intervention, and WASH between baseline and endline needed to be presented in a separate table.

Response: Given the short follow-up period (9 months) we did not ask about socio-economic characteristics at endline as we would not expect these to change. We briefly describe the feasibility and acceptability of the intervention on page 12, and are preparing a separate paper which focusing in detail on the process evaluation (which includes details of the fidelity, dose, reach and acceptability of implementing the intervention). We have added a new table showing WASH at the unannounced

visits during follow-up (Table 1).

2. Knowledge questions, myth questions and menstrual cycle questions in Table 1 needed describe in the method section.

Response: We have added these to the methods (page 8).

3. Page 11, line 3-5: "Knowledge of puberty and menstruation was poor at baseline and endline, especially among boys" How do the authors say this is poor at endline?

Response: The sentence following this gives the proportion of boys who answered all 9 questions correctly, and shows that increased from 4.1% at baseline to 12.7% at endline. We view 12.7% as a poor level of knowledge, although it is better than at baseline.

• What do the AOR data means--- needed to clarify in the text?

Response: AOR is the adjusted odds ratio as defined on page 10.

• The data in Table 1, do the authors find that the changes between baseline and endline were statistically significant for both boys and girls? Needed to talk in the result section.

Response: The 95% CI for the change between knowledge for both girls and boys between baseline and endline is given on page 13. The 95%CI for both girls and boys excludes one, showing that the finding is statistically significant.

• What are the specific differences between the boys and girls not clearly described in the text in page 11 between line numbers 3-14.

Response: On page 13, we provide the proportion of boys and girls respectively who answered all 9 knowledge questions correctly, and the adjusted odds ratio for the change from baseline to endline.

• Similarly, line number 19-35 in page 11, the difference between two age group is not clearly specified from the table 2 data.

Response: Table 2 (now Table 3) shows the results by age. We have now added into the text on page 13 that the p-value for effect modification was ≥ 0.2 for each of these variables, showing no evidence of effect-modification by age and school.

• In table 2, the perception questions and effective methods needed to bring in method section.

Response: We have added these to the methods section on page 8.

• No quantitative outcome data in regards to objectives (education, health and wellbeing) are presented except school attendance as shown in Table 3.

Response: The objective mentioned by the reviewer are for a future cluster randomised trial, not the present pilot study. We have clarified the objectives for the pilot study on page 5. The quantitative outcomes in the pilot study are given in the results section and Tables 2-4.

F. Finally ---

Since this is a before and after analysis, the authors needed to bring analysis that shows the associated factors (Puberty education, drama skirt, menstrual management kit, pain management, and WASH kits) related to interventions influenced school attendance, health and well being outcomes. i.e. authors needed to do multivariate analysis by adjusting school, gender, age etc. The authors may take the hints of analysis (Table 4) from the paper "Alam MU, et al. BMJ Open 2017;7:e015508. doi:10.1136/bmjopen-2016-015508"

Response: As described in the methods, we adjusted for school, gender and age. The main analysis is looking at the impact of the intervention by comparing endline vs baseline reports of the outcomes of interest.

Reviewer 2

1. Introduction

-Considering be more specific in stating the aim of the pilot study. To pilot test a multi-component school based MHM intervention and to prepare for a future cluster-RCT is fine. But also be clear with the aim for pilot itself. The results presented flexibility acceptability and effectiveness. The writing of the paper and presentation of those results needs to be guided by study aim.

Response: We thank the reviewer for these comments and have included the specific objectives of the pilot study on page 5 at the end of the introduction

-In the introduction, the authors stated the barriers for menstrual health for low-income context including products, water, facilities. But according to literature, menstrual health management barriers not only limit to products, water, facilities. Many studies also show the perceptions, stigma, and lack of knowledge extra also influence menstrual health management. Especially this study mentioned those factor in the methods and results part, therefore, a clear introduction with literature support is recommended.

Response: We thank the reviewer for raising this point. Addressing stigma is an important part of our intervention, and is the first point mentioned when we introduce the intervention package on page 6, as well as being mentioned throughout the paper. We have also included an additional sentence on the importance of perceptions, stigma and lack of knowledge as the second sentence of the introduction (page 4).

-managing menstruation is largely influenced by cultural and contextual background. Please give some background information to help the reader understand this population and the challenges they are facing. A better understanding of their current practice and availability of menstrual health management program is needed.

Response: We thank the reviewer for this point and have added a paragraph on this on page 5.

-This study can also benefit from reviewing of the existing menstrual health programs. Actually there are a number of quasi-experimental intervention control studies published in this area with effective outcomes.

Response: We have expanded the description of the systematic reviews on MHM interventions on page 4.

Methods

-The manuscript stated that students with low social-economic status were purposively selected. 1) What the definition of low social-economic status? 2) Why limit to low-social-economic status, because everyone need menstrual health information especially during their puberty.

Response: In our formative work, we included both 'low' and 'high' socio-economic schools, and found that there was less of a problem of missing school due to menstruation in the high SES school. We have added this to the introduction (page 5). However, there were still problems with MHM in high SES schools and we agree with the reviewer that work on this topic should not be restricted to low SES schools. We have added this as a limitation to the study (page 20).

Ethical consideration wise, how this study would influence the accessibility of menstrual management of students with better social-economic status. 3) How this would affect generalizability of this study?

Response: As above - we have added this as a limitation of the study (page 20). A future trial in 30 schools is planned and could include schools with better socio-economic status.

-Intervention. (1) It is stated that "MHM intervention needs to address stigma, education, attitudes, and psychosocial well-being". Again, a better understanding of the problems from the introduction session would better support the content of intervention.

Response: We have added sections in the introduction (pages 4 and 5) to expand on the issues of stigma, attitudes and perceptions of menstruation both generally and specifically in this setting.

(2) Any measures taken to ensure fidelity as one person trained 30 teacher.

Response: A detailed process evaluation was conducted which will be published separately. This has been noted on page 9 and 19. Briefly, we included measures of fidelity e.g. a puberty training report and school action plan which reported on the fidelity of the delivery of the puberty training education sessions.

(3) The intervention talked about one person trained 30 teachers while the results measured students. There are parts missing. I presume the teachers would also teach the students. The study procedure is unclear. The dosage and format of training is unclear. Are boys and girls attending class together? If that's the case, how students take it as you mentioned stigma earlier on. You had content of parental responsibilities for supporting girls with menstruation and also collected data from parent. Did parents also participated in the study? How comfortable the girls are with presence of boys and parents?

Response: We have expanded the description of the puberty education component (page 7) to say that 3 facilitators (one from WoMena and 2 educational consultants) trained 30 teachers who were to train students in their respective school. The schools were to develop an action plan and deliver the number of sessions specified. The qualitative study showed that acceptability of these sessions was good – the only issue was that boys would have liked a boys-only session (which we will include in a further trial). We mention this in the discussion (page 19), along with mention that inclusion of boys in the education session was a key factor in addressing stigma.

Parents participated by attending the drama skit (page 7) and we held in-depth interviews with 10 parents who had attended the skit (page 9 and 12).

-Study design and evaluation. No report characteristics especially psychometric property of the instruments used. (1) selection of suitable instruments is crucial for this topic. Questionnaire used in one culture/country may not make sense for people in different culture.

Response: The relevant instrument here is the SDQ-25, and we have added a brief description of the four domains covered by this on page 8, with a reference. The SDQ has proven reliability and validity in a number of studies across Europe, Asia, Australia and South America[1-3], but a recent review of the use of the SDQ among children and adolescents in Africa highlighted that this tool has been used in 54 studies in Africa[4], but there has only been one psychometric validation of the tool which found satisfactory internal consistency [5]. We have added this as a limitation on page 20.

Questionnaire used for adults may not be suitable for adolescent as their menstruation characteristics differs. (2) A clear operational definition is also recommended as to help reader understand what the instruments trying to measure.

Response: We have added this in the introduction (page 4)

-power analysis is needed in determine the sample size.

Response: The sample size of is was a pilot study and we have clarified the objectives (now listed on page 5). However, we had good power to detect differences in binary outcomes between baseline and endline and have added a sentence on the power on page 10.

-who, when, how the qualitative interview is conducted? Any ethical consideration for interview adolescents. Students and adolescents belong to vulnerable group which requires more careful ethical consideration. Please add one session taking about ethical issue.

Response: We have expanded the description of ethical issues on page 6, including the details of qualitative data collection. Students and teachers were interviewed at their respective schools while parents chose a place of their convenience.

-Patient and public consideration. There are no patients for participation of this study. What is the role of this part in your intervention?

Response: This is a standard section for BMJ Open, but we agree with the reviewer that this is not appropriate, and have reworded to "Participant and public consideration"

Results

- Feasibility and acceptability of implementing the intervention. (1) Is this intervention a mandatory course or students choice for free? I am not sure how this is organised, therefore, the feasibility is unclear.

Response: The intervention was not mandatory, participation of the schools, students and teachers was voluntary. We have clarified this on page 6, and the first line of the results (page 11) shows the number of eligible participants who consented to participate (95%).

(2) The first paragraph talked about how this program enabled some girls to talk about menstruation to their parents especially father while the quote is from the parents side saying they started talking to their child.

Response: We have added a quote from the girls on page 12 which mentions their increased comfort in discussing menstruation with their parents.

-Not sure how appropriate the boys can answer some of the questionnaire as the definition and property of questionnaire is missing. Seems SDQ measures behavior.

Response: The boys were asked the questions on knowledge of puberty and menstruation, perceptions and myths, and the SDQ. The questionnaires will be available as part of our data sharing strategy.

-How the qualitative and quantitative data converged. I mean for example, knowledge improvement in terms of increasing in knowledge questionnaire score can be reflected by their statement of improved understanding of menstruation.

Response: We agree that the qualitative and quantitative data support each other, and have stated this by showing both results throughout the paper, and in the discussion.

-A clear statement of study objective may help structuring the results to improve clarity.

Response: We have expanded our paragraph on the objectives on page 5.

-Again, there are many intervention studies with control group using menstrual health programs to improve menstrual health as well as public health initiatives to improve hygienic facilities. BMJ open is an international study, consider the implications this study has for global readers. What are the crucial elements identified as to move forward the menstrual health management programs. Involving boys and parents can be one, but it may need very clear rationale and description to make their role explicit.

Response: We agree this is an important point, and the 4 points that are innovative in MENISCUS, are also likely to be relevant to other settings. We have clarified this in the discussion (pages 21-22).

[1] W. Woerner, B. Fleitlich-Bilyk, R. Martinussen, J. Fletcher, G. Cucchiaro, P. Dalgarrondo, M. Lui, R. Tannock, The Strengths and Difficulties Questionnaire overseas: evaluations and applications of the SDQ beyond Europe, *Eur Child Adolesc Psychiatry* 13 Suppl 2 (2004) 1147-54.

[2] P. Vostanis, Strengths and Difficulties Questionnaire: research and clinical applications, *Curr Opin Psychiatry* 19(4) (2006) 367-72.

[3] R. Goodman, The Strengths and Difficulties Questionnaire: a research note, *J Child Psychol Psychiatry* 38(5) (1997) 581-6.

[4] N. Hoosen, E.L. Davids, P.J. de Vries, M. Shung-King, The Strengths and Difficulties

Questionnaire (SDQ) in Africa: a scoping review of its application and validation, Child Adolesc Psychiatry Ment Health 12 (2018) 6.

[5] E. Kashala, I. Elgen, K. Sommerfelt, T. Tylleskar, Teacher ratings of mental health among school children in Kinshasa, Democratic Republic of Congo, Eur Child Adolesc Psychiatry 14(4) (2005) 208-15.

VERSION 2 – REVIEW

REVIEWER	Amal K Halder Oxfam GB, South Sudan
REVIEW RETURNED	02-Nov-2019

GENERAL COMMENTS	30 October 2019 Reviewer: Amal Halder Abstract: Result:  Pg#2; Line 45-48: The authors say “The intervention was acceptable and feasible to deliver, and there were substantial reported improvements in MHH.”... this is a conclusion. Here needs result of how it indicates that the intervention was acceptable and feasible to deliver! Pg#2; line#54-57: Authors mentioned “The diary data and qualitative data indicated a potential impact of the intervention on improving menstrual-related school absenteeism.”... Need to present briefly what was the diary data and qualitative data exactly. Conclusion Pg#3; line#7-10: Mentioned “A cluster-randomised trial is needed to evaluate rigorously the intervention effects on MHM and school attendance.”... How authors can recommend this without data! I meant to say, what is the justification or what was the limitation of this study design so that RCT could be needed! Methods Study setting and participants Page#6; Line 13-18: “Low SES schools are characterized by parents with low income, lower education achievement and under resourced in terms of low grade of teachers and facilities such as infrastructure.”... there is scope to specify more about this statement. Data management and analysis Authors calculated adjusted odds ratios (AOR), adjusted mean differences (AMD) and 95% confidence intervals (CI). Authors used random-effects logistic regression accounting for within-individual clustering. However, they forgotten to explain what were the factors for which ORs were adjustment, which is essential to mention; although adjusting factors is mentioned for fixed effect model. Authors also need to explain the AOR, AMD and CI for the non-statistical audience in non-statistical form i.e. what exactly AOR and CI indicate or mean for non-statisticians. Finally, being a statistician I would interestingly want to see the stata codes used for the random-effects logistic regression model and fixed effect logistic regression models in calculating AORs. Simply they can send those codes to my email box. Also besides AOR, authors need to insert p-values besides CIs to understand whether the differences were statistically significant. It will be most convenient for readers to understand data in case
--

	AORs are replaced simply by prevalence difference (PD) alongwith CIs and p-values. 7. In regards to presentation of AOR or PD, authors may take help from the Table 4 in the article linked here https://bmjopen.bmj.com/content/7/7/e015508 ; however, keep in mind that your study is baseline versus endline which is different. 8. Objective and outcome in regards to sample size estimation is out of space in this study. It's a methodological aspect. This study does not present any result in regards to sample size estimation. Results 9. In Table 2; please insert the numbers in bracket beside % figures. 10. Pg#13; Line#52-55: "There was no evidence of moderation by age (Table 4; $p>0.2$ for effect modification for each variable) or school ($p>0.1$ for effect modification for each variable; results not shown)." ... Do not understand what does this mean and how does this link with other data in line#39-52. 11. Pg#14; Table 3: Do not understand the indicator "Knowledge excluding fertility question". Please rephrase the write up. 12. Pg 15, line 26-28: Authors say "using manufactured menstrual materials exclusively during their LMP increased from 73.0% at baseline to 89.1% at endline (AOR=4.14, 95%CI 2.04-8.40)"... I have strong doubt about AOR calculations, the difference can not be 4 times increase. Need to check the stata code. 13. Pg 15, line 39-41: Authors say "There was no evidence of effect-moderation by age or school (Table 4)". This does not make sense unless the authors presents unadjusted and adjusted OR side by side. Currently in Table 4, no such data is presented. 14. Pg 15 47-60 and pg 16, line 1-20: There a total confusion in presentation of data. For example – 1) There was no statistical change in regards to complain of pain during LMP between baseline and endline. 2) using painkillers during their LMP increased from 46.7% at baseline to 60.8% at endline (not sure whether the difference is statistically significant or not, need to see p-value), and 3) "The qualitative findings confirmed that non-pharmacological methods of pain relief were popular and effective"..... these are contradictory really. 15. Table 5 needs p-values too to understand the statistical significant differences. Discussions 16. Pg#19: Starting discussions point may not true. There are other studies too where girls were provided menstrual hygiene kits plus other WASH interventions. References -- https://bmcpublikealth.biomedcentral.com/articles/10.1186/s12889-018-6360-2 https://www.jstor.org/stable/24686633 and UNICEFs WASH in Schools (WinS) programmes are examples. However, I think there some other important findings could be focused first based on objective of the study. 17. Pg 19, line 10-14: "The improved school attendance during menstruation in this study was attributed to improved pain management, knowledge to track menstrual cycles, and the provision of reusable pads in qualitative interviews."... this has no evidence from the data yet unless multivariate analysis is done. The authors did not do any such multivariate analysis in this regard although the data has scope to do that. 18. Currently, the discussions are mixed with recommendations and conclusions. I recommend authors should focus on discussions based on potential findings from the data in result section. Followed by study limitations and defense in results against limitations. Finally presents conclusions and recommendations based on discussions.
--	--

REVIEWER	Jing Jing Su The Chinese University of Hong Kong
REVIEW RETURNED	28-Oct-2019

GENERAL COMMENTS	 1. Please match the study objectives with the results (the results should be presented clearly to fully answer your research objective/questions, for example, you presented effectiveness in terms of knowledge, attitude, practice, anxiety and depression but never mentioned in objective). The objectives should also guide the discussion so that its easier for the readers to follow. 2. Please be consistent in stating the aims and objectives of the study in abstract and main body. 3. Please provide more details of intervention so that people can replicate (e.g. dosage, format) 4. Please present name and psychometric property of questionnaires been used 5. Please provide information about trustworthiness of the qualitative results 6. Please provides ethics regarding obtain informed consent 7. Please check the definition of longitudinal cohort study. Seems inappropriate to describe your study as longitudinal cohort study in Page 9 study design and evaluation. 8. Please organize the presentation to highlight the key findings and recommendations
------------------	---

VERSION 2 – AUTHOR RESPONSE

Reviewer: 1

Reviewer Name: Amal K Halder

Institution and Country: Oxfam GB, South Sudan Please state any competing interests or state 'None declared': None

Please leave your comments for the authors below

30 October 2019

Reviewer: Amal Halder

Abstract:

Result:

1. Pg#2; Line 45-48: The authors say “The intervention was acceptable and feasible to deliver, and there were substantial reported improvements in MHH.”... this is a conclusion. Here needs result of how it indicates that the intervention was acceptable and feasible to deliver!

Response: We have altered the first line of the results in the abstract as follows “There were high levels of uptake of the individual and behavioural intervention components (puberty education, drama skit, MHM kit, and pain management).” Due to word constraints of the abstract (max 300 words) we are restricted in the detail we can provide here.

2. Pg#2; line#54-57: Authors mentioned “The diary data and qualitative data indicated a potential impact of the intervention on improving menstrual-related school absenteeism.”... Need to present briefly what was the diary data and qualitative data exactly.

Response: As above, we cannot provide details in the abstract, but these results are given in the main body of the paper (pages 17-19).

Conclusion

3. Pg#3; line#7-10: Mentioned “A cluster-randomised trial is needed to evaluate rigorously the intervention effects on MHM and school attendance.”... How authors can recommend this without data! I meant to say, what is the justification or what was the limitation of this study design so that RCT could be needed!

Response: We provide the justification for this as the last bullet point of “strengths and limitations” section (page 3 lines 18-20) and in the discussion (pages 22 lines 9-19)

Methods

Study setting and participants

4. Page#6; Line 13-18: “Low SES schools are characterized by parents with low income, lower education achievement and under resourced in terms of low grade of teachers and facilities such as infrastructure.”... there is scope to specify more about this statement.

Response: We have added an additional sentence as follows “In these schools, poor student-teacher relationship and personal hygiene were common, and are key barriers to good MHM practices and academic performance.” (page 6 lines 8-10)

Data management and analysis

5. Authors calculated adjusted odds ratios (AOR), adjusted mean differences (AMD) and 95% confidence intervals (CI). Authors used random-effects logistic regression accounting for within-individual clustering. However, they forgotten to explain what were the factors for which ORs were adjustment, which is essential to mention; although adjusting factors is mentioned for fixed effect model. Authors also need to explain the AOR, AMD and CI for the non-statistical audience in non-statistical form i.e. what exactly AOR and CI indicate or mean for non-statisticians. Finally, being a statistician I would interestingly want to see the stata codes used for the random-effects logistic regression model and fixed effect logistic regression models in calculating AORs. Simply they can send those codes to my email box.

Response: We state on page 10 line 24-25 that analyses are adjusted for school, age (<16 and ≥16 years) and gender where appropriate (i.e. for analyses including both genders). Odds ratios and confidence intervals are standard in the epidemiological literature so we have not explained meaning (and no longer present odds ratio – see below). However we can do so if the Editors request this.

The stata code was standard for random-effects logistic regression, for example, xtlogit `x' i.study i.schcode i.age_cat, i(idno) or where study = baseline or endline, schcode=school, age_cat = age group

6. Also besides AOR, authors need to insert p-values besides CIs to understand whether the differences were statistically significant. It will be most convenient for readers to understand data in case AORs are replaced simply by prevalence difference (PD) alongwith CIs and p-values.

Response: We have added in p-values throughout, and we have replaced the odds ratios with prevalence ratios (PR) and prevalence differences (PD) throughout the paper.

7. In regards to presentation of AOR or PD, authors may take help from the Table 4 in the article linked here <https://bmjopen.bmj.com/content/7/7/e015508> ; however, keep in mind that your study is baseline versus endline which is different.

Response: We follow this format and show the adjusted prevalence ratio, the adjusted prevalence difference, and the p-values in Tables 3-5 and the text.

8. Objective and outcome in regards to sample size estimation is out of space in this study. It's a methodological aspect. This study does not present any result in regards to sample size estimation.

Response: The power calculation is given on page 10 lines 9-12. “The planned sample size of 200 girls (and 200 boys, respectively) provides 85% power to detect an odds ratio of two between baseline and endline assuming baseline prevalence of 20%, allowing for within-individual correlation (intracluster-correlation=0.05).

Results

9. In Table 2; please insert the numbers in bracket beside % figures.

These have now been included in Table 2.

10. Pg#13; Line#52-55: “There was no evidence of moderation by age (Table 4; $p>0.2$ for effect modification for each variable) or school ($p>0.1$ for effect modification for each variable; results not shown).”... Do not understand what does this mean and how does this link with other data in line#39-52.

Response: We have clarified this by using the term intervention effect modification on page 14 lines 6-8: “There was no evidence that the intervention effect differed by age (Table 4; $p \geq 0.2$ for effect modification for each variable) or school ($p \geq 0.1$ for effect modification for each variable; results not shown).“

11. Pg#14; Table 3: Do not understand the indicator “Knowledge excluding fertility question”. Please rephrase the write up.

Response: We have clarified by using the wording: “8 knowledge questions (excluding fertility question) correct” in Table 3

12. Pg 15, line 26-28: Authors say “using manufactured menstrual materials exclusively during their LMP increased from 73.0% at baseline to 89.1% at endline (AOR=4.14, 95%CI 2.04-8.40)”... I have strong doubt about AOR calculations, the difference can not be 4 times increase. Need to check the stata code.

Response: The odds ratio is correct, but given the high prevalence it is not a close estimate of the adjusted prevalence ratio. In line with comments above to show prevalence difference, we now show adjusted prevalence ratios throughout the paper, rather than adjusted odds ratios, which addresses this issue.

13. Pg 15, line 39-41: Authors say “There was no evidence of effect-moderation by age or school (Table 4)”. This does not make sense unless the authors presents unadjusted and adjusted OR side by side. Currently in Table 4, no such data is presented.

Response: Comparison of unadjusted and adjusted ORs informs on the presence of confounding but not effect-modification.

14. Pg 15 47-60 and pg 16, line 1-20: There a total confusion in presentation of data. For example – 1) There was no statistical change in regards to complain of pain during LMP between baseline and endline. 2) using painkillers during their LMP increased from 46.7% at baseline to 60.8% at endline (not sure whether the difference is statistically significant or not, need to see p-value), and 3) “The qualitative findings confirmed that non-pharmacological methods of pain relief were popular and effective”..... these are contradictory really.

Response: Each of these statements if correct – as follows:

1) There was no statistical change in regards to complain of pain during LMP between baseline and endline.

This is correct – as stated on page 16 lines 23-24 “Most girls reported pain during menstruation at both baseline and endline (74.3% and 71.0%, respectively).”

2) using painkillers during their LMP increased from 46.7% at baseline to 60.8% at endline (not sure whether the difference is statistically significant or not, need to see p-value)

This is correct, and the 95%CI shows that the difference is “statistically significant” (CI APR=1.26, 95%CI 1.03-1.55, $p=0.03$) (we now show the p-value)

3) “The qualitative findings confirmed that non-pharmacological methods of pain relief were popular and effective”..... these are contradictory really.

This statement is also correct as use of both non-pharmacological and pharmacological methods of pain relief increased. The proportion of girls with pain at LMP who reported using NO method decreased from 25% to 6% among those aged <16 years, and 20% to 7.5% among those aged ≥ 16 years.

15. Table 5 needs p-values too to understand the statistical significant differences.

Response: These have been added

Discussions

16. Pg#19: Starting discussions point may not true. There are other studies too where girls were provided menstrual hygiene kits plus other WASH interventions. References --

<https://bmcpublichealth.biomedcentral.com/articles/10.1186/s12889-018-6360-2>

<https://www.jstor.org/stable/24686633> and UNICEFs WASH in Schools (WinS) programmes are examples. However, I think there some other important findings could be focused first based on objective of the study.

Response: We thank the reviewer for this point. However, the references given here do not give results of interventions but are descriptive studies, or call for a multi-component intervention. Some other studies do include some education component as well as pad

provision, for example, but to our knowledge as is the first which has been planned as a multi-component intervention without one major component taking precedence.

17. Pg 19, line 10-14: "The improved school attendance during menstruation in this study was attributed to improved pain management, knowledge to track menstrual cycles, and the provision of reusable pads in qualitative interviews."... this has no evidence from the data yet unless multivariate analysis is done. The authors did not do any such multivariate analysis in this regard although the data has scope to do that.

Response: We have clarified this sentence as the evidence cited is from the qualitative interviews. This now reads (page 22, lines 23-25): "This aligns with data from qualitative interviews, in which girls attributed the improved school attendance during menstruation to improved pain management, knowledge to track menstrual cycles, and the provision of reusable pads. "

18. Currently, the discussions are mixed with recommendations and conclusions. I recommend authors should focus on discussions based on potential findings from the data in result section. Followed by study limitations and defense in results against limitations. Finally presents conclusions and recommendations based on discussions.

Response: We thank the reviewer for this point and have revised the discussion as suggested (also to address the comments from Reviewer 1).

Reviewer: 2

Reviewer Name: Jing Jing Su

Institution and Country: The Chinese University of Hong Kong Please state any competing interests or state 'None declared': None

Please leave your comments for the authors below

1. Please match the study objectives with the results (the results should be presented clearly to fully answer your research objective/questions, for example, you presented effectiveness in terms of knowledge, attitude, practice, anxiety and depression but never mentioned in objective). The objectives should also guide the discussion so that its easier for the readers to follow.

Response: We thank the reviewer for this point. The previous objectives referred to the pilot study as a whole rather than results presented in this manuscript. We have rewritten the objectives to match the results given in this paper (Page 5 lines 19-25)

"The objectives of the pilot study are to: i) Assess the feasibility and acceptability of implementing a combined package of MHH intervention elements developed in the formative work, delivered to a whole secondary school year for nine months; ii) Assess the outcomes of the intervention package at baseline and endline (knowledge and attitudes towards puberty, menstruation and pain management); iii) Pilot the use of daily diaries to estimate school attendance, and compare attendance with estimates using registers, observation visits and retrospective self-report; iv) Estimate school retention over a 9 month period."

2. Please be consistent in stating the aims and objectives of the study in abstract and main body.

Response: Due to word constraints in the abstract (max 300 words) we have a shortened version of the aim in the abstract but it is very similar to the fuller version in the main body.

The objectives expand on the aim

Abstract aim (page 2 lines 3-5): "The aim of this study was to pilot-test an intervention to improve MHH and school attendance in Uganda, in preparation for a future cluster-randomised trial."

Main body aim (page 5 lines 6-9): The aim of this study was to pilot-test a multi-component school-based MHH intervention ("MENISCUS") and to prepare for a future cluster-randomised trial (CRT) which will evaluate the impact of the intervention on secondary school attendance, performance, menstruation knowledge, health and wellbeing outcomes in Uganda."

(Objectives as above in response to point 1)

3. Please provide more details of intervention so that people can replicate (e.g. dosage, format)

Response: This is now included as Supplementary Table 1 (referred to on page 7, line 6)

4. Please present name and psychometric property of questionnaires been used

Response: We provide the tool used for psychosocial wellbeing on page 8 (lines 8-12) and have added additional references which give the psychometric properties. We also refer to the properties of the SDQ in the discussion (page 21, lines 23-26). Other questions are given in Table 1.

5. Please provide information about trustworthiness of the qualitative results

Response: The trustworthiness of the qualitative data was assessed in accordance with the approaches suggested by Krefting (1991: 217) to ensure credibility, applicability, dependability and confirmability of the data. The training, skill and careful supervision and support of researchers in the conduct of qualitative data collection underpins this approach, as we have argued elsewhere (Rutakumwa 2019). We have added this on page 9 (lines 14-18).

6. Please provides ethics regarding obtain informed consent

Response: As stated on page 28, ethical approval was received from UVRI Research Ethics Committee, the Uganda National Council for Science and Technology (UNCST) and the London School of Hygiene and Tropical Medicine. The procedures used to obtain informed consent are given on page 6, lines 11-16

“Written informed consent was sought from students aged ≥ 18 years, and from the parents/caretakers of those aged < 18 years, with student assent. We also received written informed consent from teachers and parents/caretakers participating in interviews. Participation was voluntary, and confidentiality was ensured by conducting interviews in a private setting and keeping the data collected on secure servers without identifying information.”

7. Please check the definition of longitudinal cohort study. Seems inappropriate to describe your study as longitudinal cohort study in Page 9 study design and evaluation.

Response: In the abstract, we describe the design as “Longitudinal study with pre-post evaluation of a pilot intervention” and we have used this wording in the main body now too (page 8 line 3). This is also the design described in the abstract (page 2 line 6)

8. Please organize the presentation to highlight the key findings and recommendations

Response: We have re-structured the discussion to follow the 3 objectives, with each section giving key findings. We then give key recommendations at the end of the discussion (pages 22-23).

VERSION 3 – REVIEW

REVIEWER	Amal Krishna Halder Oxfam GB, South Sudan
REVIEW RETURNED	01-Jan-2020
GENERAL COMMENTS	The manuscript improved a lot. I think this could be accepted now for publications in case write up has clarity and standard. My review was mostly based on statistical and this part is mostly done. Thanks